# Do Vision-Language Models Rotate in Mind? Evaluating Spatial Transformation Reasoning

## Abstract

Vision-language models (VLMs) have achieved impressive performance across diverse tasks, yet their ability to mentally perform spatial transformations—rotating, translating, and manipulating objects—remains poorly understood. We present **TransformEval**, a systematic benchmark that evaluates spatial transformation reasoning across 2D shapes, 3D mental rotation, and multi-object scenes, uniquely distinguishing between *state prediction* (determining outcomes after transformations) and *transformation inference* (recovering operations from state changes). Our evaluation of state-of-the-art VLMs reveals fundamental limitations: models struggle with basic transformations that humans solve effortlessly, consistently perform better at inferring transformations than predicting their outcomes—opposite to human cognitive patterns—and frequently fail at translation operations and transformation ordering. These findings suggest that VLMs rely on pattern matching rather than mental simulation, lacking the spatial reasoning capabilities necessary for applications in robotics, augmented reality, and other domains requiring genuine spatial intelligence. Our benchmark provides a framework for measuring progress toward human-like spatial understanding in vision-language models.

## 1 Introduction

Spatial transformation reasoning—the ability to mentally manipulate objects through rotation, translation, and other geometric operations—represents a cornerstone of human intelligence. This capability enables us to pack luggage efficiently, navigate using maps, assemble furniture from diagrams, and imagine how objects appear from different viewpoints. As Shepard and Metzler's (Shepard & Metzler, 1971) seminal work demonstrated, humans perform mental rotations through analog processes that mirror physical transformations, with reaction times proportional to rotation angles. This fundamental cognitive ability, which even young children develop through play, reveals how deeply embedded spatial transformation is in human cognition.

Yet despite remarkable achievements in image analysis, mathematical reasoning, and visual question answering, vision-language models (VLMs) conspicuously lack this basic capability (OpenAI et al., 2024; Reid et al., 2024; Yue et al., 2024). The same models that can solve complex physics problems cannot predict how a simple L-shaped block will look after a 90-degree rotation. This gap between sophisticated symbolic reasoning—where VLMs excel—and basic spatial transformation—where they fail catastrophically—exposes a fundamental limitation in current architectures.

The inability to perform mental transformations has profound implications for real-world deployment. Robotic systems using VLMs for scene understanding cannot reliably predict object states after manipulation (Chen et al., 2024a). Augmented reality applications struggle to maintain spatial coherence as users move through environments (Cheng et al., 2024). Navigation systems fail to anticipate how spatial relationships change with movement. Without the analog-like mental rotation capabilities that Shepard and Metzler identified in humans, these systems remain limited to static scene interpretation rather than dynamic spatial reasoning.

Recent benchmarks have documented VLMs' spatial limitations (Ramakrishnan et al., 2025; Wang et al., 2024), revealing poor performance on tasks requiring metric understanding, frame of refer-

ence disambiguation, and 3D reasoning. However, these evaluations primarily assess static spatial relationships: whether objects are left or right, near or far, occluded or visible. They miss the crucial dimension that Shepard and Metzler's paradigm exposed—the ability to mentally simulate continuous transformations. Understanding that "the cup is left of the plate" differs fundamentally from predicting where the cup will be after rotating the table 90 degrees.

This distinction matters because real-world spatial reasoning is inherently dynamic. Maps must be mentally rotated to align with one's orientation. Assembly instructions require imagining how parts fit together through sequential rotations and translations. Even simple tasks like packing a suitcase demand mental simulation of how objects can be transformed to fit available space. The analog mental rotation process that humans use naturally—where we mentally "watch" objects rotate through intermediate positions—appears entirely absent in current VLMs.

We introduce **TransformEval**, a comprehensive benchmark specifically designed to evaluate spatial transformation reasoning in VLMs, inspired by classic paradigms from cognitive psychology. Unlike existing benchmarks that test static spatial knowledge, TransformEval systematically probes the ability to mentally manipulate objects through rotation, translation, and their compositions. Our framework makes three key contributions:

1. **Bidirectional evaluation paradigm**: We distinguish between forward reasoning (predicting states after transformations) and inverse reasoning (inferring transformations from state changes), revealing that VLMs exhibit opposite patterns to human cognition—performing better at transformation inference than state prediction, contrary to the natural human tendency for forward mental simulation.

2. **Hierarchical complexity framework**: Our benchmark progresses systematically from 2D shape transformations to 3D mental rotation (directly inspired by Shepard and Metzler's paradigm) to multi-object scene manipulation, with carefully controlled distractors that diagnose specific failure modes.

3. **Compositional transformation analysis**: We specifically evaluate sequential transformations where order matters—a critical aspect of real-world spatial reasoning where $T_1 \circ T_2 \neq T_2 \circ T_1$—finding that models catastrophically fail when transformations must be composed.

Our evaluation of state-of-the-art VLMs reveals fundamental limitations in spatial transformation reasoning. Models achieve below 40% accuracy on basic 2D rotations that humans solve well, with performance degrading further on 3D mental rotation tasks (30%) that directly parallel Shepard and Metzler's original experiments. The surprising finding that models perform better at inferring transformations (50%) than predicting their outcomes (27%) suggests reliance on pattern matching rather than the analog mental simulation processes that characterize human spatial cognition.

These findings have immediate practical implications. Applications requiring spatial transformation understanding—from robotic manipulation to architectural design tools—cannot rely on current VLMs for the kind of mental rotation abilities humans use effortlessly. Our error analysis reveals specific bottlenecks: translation operations are disproportionately challenging (47% of errors), transformation order is frequently ignored (32% of errors), and compositional reasoning shows catastrophic failure. By identifying these systematic weaknesses and contrasting them with well-established human cognitive processes, TransformEval provides a roadmap for developing models with genuine spatial transformation capabilities, ultimately enabling more robust spatial intelligence in vision-language systems.

## 2 RELATED WORK

### 2.1 SPATIAL REASONING IN VISION-LANGUAGE MODELS

Recent advances in Vision-Language Models (VLMs) and Large Language Models (LLMs) have led to a proliferation of benchmarks and methodologies aimed at evaluating their spatial reasoning capabilities. Early works such as Sampat et al. (2024) established foundational tasks for assessing static spatial reasoning by focusing on object relationships and hypothetical actions over images. These

studies provided critical insights into how VLMs interpret and reason about spatial configurations when the challenges are confined to static scenes.

Building on these efforts, several recent studies have turned their attention to dynamic spatial transformations—tasks that require models to process sequential actions like rotation and translation. For example, Rizvi et al. (2024) characterizes the spatial reasoning abilities of LLMs with tasks designed to capture the sequential nature of transformations. Similarly, works such as the CLEVR Mental Rotation Tests Beckham et al. (2022) and Wu et al. (2024) highlight that while models perform well on static benchmarks, they often struggle with anticipating future states and maintaining consistency when multiple, ordered transformations are applied.

Complementing these investigations, several benchmarks and datasets have been introduced to address embodied and multi-view spatial reasoning. Benchmarks such as Du et al. (2024) and Jin et al. (2023) extend evaluation into the realm of long-horizon decision making and embodied tasks, while datasets like Han et al. (2020) and Xu et al. (2022) challenge models to integrate 3D spatial reasoning with multi-view consistency.

Recent works have also explored the integration of spatial reasoning within the model architecture itself. Chen et al. (2024b) and Li et al. (2025) demonstrate promising approaches by explicitly endowing models with components to handle transformation prediction tasks. Moreover, research adopting psychometric perspectives Xu et al. (2025); Newcombe (2024) provides a complementary view by analyzing these models' fundamental spatial abilities through the lens of cognitive science.

Other notable contributions include studies such as Li et al. (2024a) and Tang et al. (2024), which focus on interpreting spatial representations from a top-view or composite spatial reasoning perspective, and Ray et al. (2024) that offers training benchmarks specifically tailored for enhancing spatial aptitude in multimodal settings.

Collectively, these works underscore a significant trend: while static spatial reasoning has been extensively studied, dynamic and sequential spatial transformations remain a challenging frontier. Our work aims to fill this gap by introducing an evaluation framework that specifically targets the sequential aspects of spatial transformations, thereby offering deeper insights into the strengths and limitations of current VLMs and LLMs.

## 3 TRANSFORMEVAL BENCHMARK

### 3.1 DESIGN PRINCIPLES

Our benchmark design follows three key principles derived from cognitive science and practical applications:

**Bidirectional Evaluation:** For each transformation scenario, we evaluate both forward (state prediction) and inverse (transformation inference) reasoning. This bidirectional approach reveals whether models possess true understanding or merely exploit statistical regularities.

**Compositional Complexity:** We systematically vary transformation complexity with compound transformations where order matters. This allows us to identify at what point model capabilities break down.

**Controlled Distractors:** Each multiple-choice question includes carefully designed distractors representing specific error types (e.g., missing translation, incorrect rotation angle, wrong operation order), enabling detailed error analysis.

### 3.2 TASK DESCRIPTIONS

#### 3.2.1 2D SHAPE TRANSFORMATIONS

We begin with transformations of distinctive 2D shapes (L, T, F, Z, arrows) rendered on a grid. Each trial involves:

- **Initial state:** A shape positioned on a 10×10 grid

- **Transformation:** Compound operation (e.g., "Rotate 315° clockwise, then translate 3 units right")
- **Task:** Select the correct final state from 4 candidates

Transformations include rotations at 45° intervals and translations of 1-3 grid units. Distractors represent common errors:

1. Missing translation (rotation only)
2. Opposite translation direction
3. Incorrect rotation angle

This task tests basic transformation capabilities while remaining simple enough for detailed error analysis. The use of non-rotationally-symmetric shapes ensures models cannot succeed through template matching.

### 3.2.2 3D MENTAL ROTATION

Following classic mental rotation paradigms Shepard & Metzler (1971), we use distinctive 3D objects composed of connected cubes:

- **Objects:** Self-avoiding chains of 10-20 cubes, ensuring unique viewing angles
- **Transformations:** Sequential rotations about principal axes (e.g., "90° about X-axis, then 180° about Z-axis")
- **Distractors:** Wrong rotation order, incorrect angles, wrong axes

This task specifically probes whether models can perform mental rotations analogous to human spatial processing. The use of compound rotations where order matters (since rotations in 3D don't commute) tests compositional understanding.

### 3.2.3 MULTI-OBJECT SCENE TRANSFORMATIONS

Our most challenging task involves transforming objects within complex scenes:

- **Scenes:** 3-10 objects of varying shapes, colors, and materials
- **Target specification:** Natural language description (e.g., "the yellow bus")
- **Transformation:** Applied to specified object while others remain fixed
- **Challenges:** Occlusion, object identification, relative position changes

This task mirrors real-world scenarios where transformations occur in cluttered environments. Models must identify the target object, apply transformations correctly, and understand how the transformation affects relationships with other objects.

## 3.3 DATASET CONSTRUCTION

We generate datasets using controlled rendering pipelines:

**2D shapes:** 300 examples using matplotlib with consistent styling, ensuring uniform difficulty across shape types.

**3D objects:** 300 examples rendered in Blender using procedurally generated distinctive objects, with controlled lighting and viewing angles.

**Scene transformations:** 300 examples using Super-CLEVRLi et al. (2023) style scenes with physics-based rendering for realistic appearance.

Each dataset is balanced across transformation types and complexities. We ensure that random guessing yields 25% accuracy by using 4-way multiple choice throughout. The locations of the correct choices among four candidates are randomized to avoid the positional bias within vision language models.

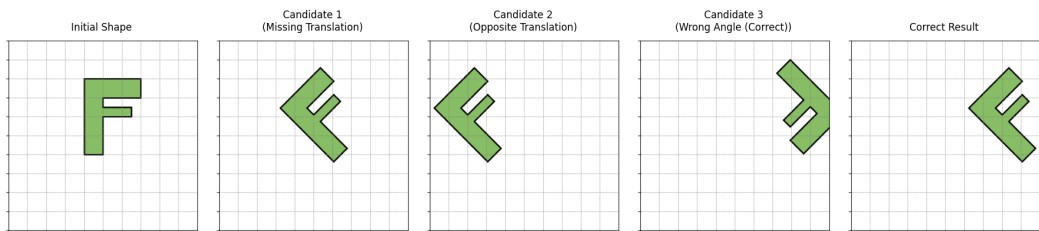

Figure 1: Example from our 2D shape transformation task. The operation is "Rotate 315° clockwise about center, then translate by 3 grids right."

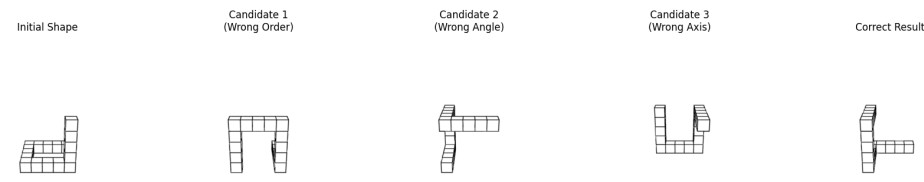

Figure 2: Example from our 3D mental rotation task. The operation is "Rotate 90° about Z-axis then rotate 90° about Y-axis." The model is instructed with X, Y, Z axes and rotation directions.

## 4 EXPERIMENTAL SETUP

### 4.1 MODELS EVALUATED

We evaluate state-of-the-art vision-language models: **Gemini-2.5-Pro** Comanici et al. (2025), **GPT-4o** OpenAI et al. (2024), **InternVL2-8B/40B** Chen et al. (2024c), **LLaVA-NeXT** Liu et al. (2024), and **LLaVA-OneVision** Li et al. (2024b).

### 4.2 EVALUATION PROTOCOL

For each example, we provide:

1. The initial state (image)
2. Transformation description (text)
3. Four candidate final states (images)

Models select the correct option and optionally provide reasoning. We use consistent prompting across models:

```
"Given the initial shape/object and the transformation described,
which of the four options (A, B, C, D) shows the correct final
state?"
```

For transformation inference tasks, we reverse the setup: providing initial and final states, asking models to identify the transformation sequence.

### 4.3 HUMAN PERFORMANCE BASELINE

We collected human performance data from participants with normal vision. Participants completed examples from each task category after brief familiarization. Human accuracy serves as an upper bound, though we note that humans also show systematic variation in mental rotation ability Voyer et al. (1995).

Figure 3: Example from our scene transformations task. The operation is "Operate on the yellow bus. Rotate 45° counterclockwise about center, then translate in the further direction."

| Model | 2D Shapes | | 3D Rotation | | Multi-Object | |
|---|---|---|---|---|---|---|
| | State | Trans. | State | Trans. | State | Trans. |
| Human | 95.2 | 89.8 | 89.4 | 75.1 | 91.6 | 80.3 |
| GPT-4o | 28.7 | 60.1 | 24.7 | 43.1 | 18.5 | 22.6 |
| Gemini-2.5-Pro | **47.2** | **71.3** | **29.8** | **54.9** | **25.1** | **33.7** |
| InternVL2-8B | 34.2 | 41.8 | 22.9 | 30.8 | 22.3 | 24.3 |
| InternVL2-40B | 36.2 | 45.2 | 25.7 | 41.3 | 24.8 | 32.7 |
| LLaVA-NeXT | 32.8 | 39.1 | 26.8 | 46.1 | 23.9 | 26.8 |
| LLaVA-OneVision | 31.0 | 37.4 | 24.9 | 44.5 | 24.1 | 28.9 |
| Random | 25.0 | 25.0 | 25.0 | 25.0 | 25.0 | 25.0 |

Table 1: Model accuracy (%) on state prediction and transformation inference tasks across three complexity levels. Bold indicates best model performance per column.

## 5 RESULTS

### 5.1 OVERALL PERFORMANCE

Table 1 presents model performance across all tasks. Several striking patterns emerge:

**1. Severe performance degradation:** All models perform far below human levels, with the best model (Gemini-2.5-Pro) achieving only 47% accuracy on 2D shape state prediction compared to 95.2% human accuracy.

**2. Asymmetric capabilities:** Models consistently perform better on transformation inference than state prediction, suggesting they recognize transformation patterns but cannot mentally simulate their effects.

**3. Task hierarchy:** Performance decreases with task complexity: 2D shapes ¿ 3D rotation ¿ multi-object scenes, but the degradation is less severe than expected, indicating that even simple transformations challenge current models.

### 5.2 ERROR ANALYSIS

Table 2 shows the distribution of error types. Key findings include:

**Translation neglect:** In 2D tasks, 46.1% of errors involve missing the translation component entirely or opposite translation, suggesting models focus on rotation while ignoring subsequent operations.

**Order insensitivity:** In 3D tasks, 32.4% of errors involve applying rotations in the wrong order, indicating models don't understand that rotation composition is non-commutative.

**Magnitude errors:** Rotation angle errors (e.g., 90° instead of 180°) account for 22.8% of 3D errors, suggesting imprecise angle representation.

Table 2: Error Distribution Comparison between GPT-4o and Gemini across Different Tasks

| Task | Model | Correct (%) | Error Types (%) | | | |
|---|---|---|---|---|---|---|
| | | | Wrong Angle | Opposite Trans. | Missing Trans. | Wrong Order/Axis |
| **2D Tasks** | | | | | | |
| 2D - Final State | GPT-4o | 28.7 | 15.3 | 21.7 | 34.3 | – |
| | Gemini | 47.2 | 6.7 | 21.4 | 24.7 | – |
| 2D - Transform. | GPT-4o | 60.1 | 7.0 | 23.5 | 9.4 | – |
| | Gemini | 71.3 | 9.7 | 13.0 | 6.0 | – |
| **3D Tasks** | | | | | | |
| 3D - Final State | GPT-4o | 24.7 | 30.3 | – | – | 45.0 |
| | Gemini | 29.8 | 22.8 | – | – | 47.4 |
| 3D - Transform. | GPT-4o | 43.1 | 17.3 | – | – | 39.6 |
| | Gemini | 54.9 | 16.8 | – | – | 28.3 |
| **Scene Tasks** | | | | | | |
| Scene - Final State | GPT-4o | 18.5 | 25.1 | 27.7 | 28.7 | – |
| | Gemini | 25.1 | 23.7 | 22.9 | 28.3 | – |
| Scene - Transform. | GPT-4o | 22.6 | 24.3 | 19.1 | 34.0 | – |
| | Gemini | 33.7 | 25.2 | 20.4 | 20.7 | – |

# 6 ANALYSIS AND DISCUSSION

## 6.1 THE STATE-TRANSFORMATION ASYMMETRY

The consistent performance gap between transformation inference and state prediction deserves deeper analysis. This asymmetry—opposite to human patterns where forward simulation is typically easier than inverse problems—suggests fundamental differences in how VLMs and humans process spatial transformations.

**Pattern matching vs. mental simulation:** Humans appear to mentally simulate transformations through analog processes Shepard & Metzler (1971), making forward prediction natural. VLMs, lacking such simulation capabilities, may instead rely on learned associations between transformation descriptions and visual patterns. This makes the inverse problem (recognizing transformations from examples) paradoxically easier.

**Discrete vs. continuous representations:** Current VLMs process images through discrete tokens and attention mechanisms optimized for recognition rather than transformation. This architectural bias may explain why they can identify "this looks like a 90-degree rotation" but cannot generate the resulting state.

**Training data implications:** Vision-language training data typically contains static image descriptions rather than transformation sequences. Models may have learned to recognize transformation-related words without understanding their geometric meaning.

## 6.2 THE TRANSLATION PROBLEM

Our error analysis reveals that translation operations are disproportionately challenging, with nearly half of 2D errors involving missed translations. Several factors may contribute:

**Attention mechanisms:** Transformer architectures use position-invariant attention patterns that may inherently struggle with translation, which is fundamentally about position change.

**Relative vs. absolute position:** While rotations maintain the object's center of mass (a relative property), translations require tracking absolute position changes—a capability that may be under-developed in current architectures.

**Sequential processing failures:** The high rate of translation omission in compound transformations suggests models may have limited working memory for multi-step operations, forgetting earlier or later steps in the sequence.

### 6.3 IMPLICATIONS FOR SPATIAL AI

Our findings have significant implications for deploying VLMs in spatial reasoning applications:

**Robotics:** Robots using VLMs for scene understanding may struggle with tasks requiring transformation prediction, such as estimating object positions after manipulation or navigating through dynamic environments.

**AR/VR:** Applications requiring accurate spatial transformation understanding (e.g., placing virtual objects that remain stable as users move) may need specialized spatial reasoning modules rather than relying on general VLMs.

**Design and CAD:** Tools using VLMs to interpret spatial instructions or generate designs from descriptions may produce incorrect results when transformations are involved.

### 6.4 TOWARDS BETTER SPATIAL REASONING

Our results suggest several directions for improving spatial reasoning in VLMs:

**Explicit 3D representations:** Incorporating explicit 3D scene representations or neural radiance fields might provide the geometric grounding necessary for accurate transformation reasoning.

**Transformation-augmented training:** Pre-training or fine-tuning on datasets with explicit transformation sequences could help models learn proper geometric operations.

**Modular architectures:** Separating perception from transformation reasoning—perhaps through specialized geometric reasoning modules—might overcome current limitations.

**Continuous representations:** Moving beyond discrete token-based processing toward continuous representations might better support analog-like mental transformations.

## 7 LIMITATIONS AND FUTURE WORK

While TransformEval provides systematic insights into VLM spatial reasoning capabilities, several limitations should be noted:

**Synthetic data:** Our use of rendered images may not fully capture real-world complexity. Future work should evaluate on natural images with real objects undergoing transformations.

**Limited transformation types:** We focus on rotation and translation, but other transformations (scaling, shearing, reflection) also deserve investigation.

**Static evaluation:** Our benchmark uses static images of states. Interactive evaluation where models manipulate objects directly could reveal different capabilities.

**Cultural and linguistic factors:** Spatial reasoning varies across cultures and languages Majid et al. (2004). Our English-centric evaluation may not generalize globally.

Future work should address these limitations while exploring:

- Developmental trajectories: Can models learn spatial transformations through curriculum learning mirroring human cognitive development?
- Intervention studies: Does explicit geometric reasoning training improve transformation capabilities?
- Architectural innovations: What computational primitives are necessary for human-like mental rotation?

## 8 CONCLUSION

This paper presents TransformEval, a comprehensive benchmark revealing fundamental limitations in how current vision-language models handle spatial transformations. Our systematic evaluation across 2D shapes, 3D mental rotation, and complex scenes demonstrates that state-of-the-art VLMs struggle with basic transformation reasoning, achieving less than 30% accuracy on tasks humans solve effortlessly.

Three key findings emerge from our analysis:

First, models exhibit an unexpected asymmetry, performing better at inferring transformations from examples than predicting transformation outcomes. This reversal of human patterns suggests VLMs lack true mental simulation capabilities, instead relying on pattern matching.

Second, translation operations prove particularly challenging, with models frequently ignoring position changes in compound transformations. This points to fundamental limitations in how current architectures represent and update spatial positions.

Third, compositional transformation reasoning—applying multiple operations in sequence—shows catastrophic performance degradation, indicating that models cannot properly compose geometric operations.

These findings have immediate practical implications. Applications requiring robust spatial reasoning—from robotic manipulation to augmented reality—cannot yet rely on general-purpose VLMs for transformation understanding. Our error analysis provides specific guidance for improvement: models need better position tracking mechanisms, working memory for sequential operations, and training on transformation-rich datasets.

Looking forward, closing the spatial reasoning gap requires rethinking how VLMs represent and manipulate spatial information. Whether through architectural innovations, training paradigm shifts, or hybrid symbolic-neural approaches, achieving human-like spatial transformation abilities remains an open challenge. TransformEval provides a systematic framework for measuring progress toward this goal, ultimately contributing to more spatially intelligent AI systems.

By highlighting these fundamental limitations and providing tools for systematic evaluation, we hope to catalyze research toward VLMs with genuine spatial understanding—models that can not only see the world but also imagine how it changes through movement and manipulation.

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
