# OpenReview forum: "Do Vision Language Models Rotate in Mind? Evaluating Spatial Transformation Reasoning"
_ICLR.cc/2026/Conference — ICLR 2026 Conference Withdrawn Submission_

### Official Review · Reviewer_q3yr · 2025-10-31

**Soundness:** 3
**Presentation:** 2
**Contribution:** 2
**Rating:** 4
**Confidence:** 5

**Summary:**

This paper focuses onVLMs’ ability to perform mental spatial transformations (e.g., rotation, translation) by introducing TransformEval, a comprehensive benchmark designed to evaluate such reasoning across three hierarchical tasks: 2D shape transformations (using asymmetric shapes on a 10×10 grid with compound rotation/translation operations), 3D mental rotation (using cube-chain objects with sequential axis rotations), and multi-object scene transformations (cluttered scenes with targeted object manipulations). TransformEval distinguishes between state prediction and transformation inference, two directions that reveal a striking asymmetry. Results show that VLMs consistently outperform at inference than at prediction, which directly contrasts with human cognitive patterns where forward simulation is easier. Overall, VLMs perform far below humans, with frequent failures in translation operations and transformation ordering, indicating that current VLMs rely on pattern matching rather than human-like mental simulation.

**Strengths:**

Well-motivated Benchmark Design: This benchmark fills a critical gap in existing spatial reasoning evaluations. Unlike prior benchmarks focusing solely on static spatial relationships, TransformEval targets dynamic spatial transformation reasoning (e.g., mental rotation, sequential manipulation). It covers hierarchical tasks (2D shape → 3D mental rotation → multi-object scenes) and introduces a bidirectional evaluation paradigm that has state prediction vs. transformation inference, enabling precise diagnosis of VLMs’ true understanding rather than superficial pattern matching. The inclusion of controlled distractors (e.g., missed translation, wrong rotation order) further supports fine-grained error analysis.

Insightful Finding: The discovery of the "inference-prediction asymmetry" (VLMs excel at inverse transformation inference but struggle with forward state prediction, opposite to humans) reveals a fundamental difference between VLMs’ pattern-matching mechanism and human-like mental simulation.

**Weaknesses:**

The TransformEval benchmark, while targeted, suffers from excessive simplification that weakens its ability to reflect complex real-world spatial reasoning scenarios. All tasks rely on artificially rendered content: 2D tasks use simple asymmetric shapes (L/T/F) on a fixed 10×10 grid, 3D tasks use rigid "cube-chain" objects with no textural or structural complexity, and multi-object scenes are based on stylized Super-CLEVR renders (not natural images). This simplification avoids real-world noise but also means VLMs’ poor performance here may not fully translate to, or overstate, their weaknesses in practical scenarios.

Transformation types are overly narrow. The benchmark only evaluates rotation and translation, ignoring common real-world spatial operations like scaling, shearing, reflection, or non-rigid deformation (e.g., folding paper, bending a wire, which are critical for applications like CAD design or robot-assisted packaging.

Task format is static and passive. All evaluations use a 4-choice static image comparison, with no interactive or dynamic assessment (e.g., asking VLMs to simulate step-by-step transformations). This fails to capture the iterative, dynamic nature of human spatial reasoning.

The paper does not verify whether key design choices (e.g., task hierarchy, distractor types) affect results. For example, it does not test if simplifying 3D rotation to single-axis (instead of sequential multi-axis) improves VLMs’ performance, or if changing grid size in 2D tasks impacts translation error rates. Without such ablations, it is hard to confirm whether the benchmark’s complexity aligns with its goal of diagnosing spatial weaknesses, or if results are skewed by specific task parameters.

The study uses a fixed prompt template for all models but does not explore how prompt engineering (e.g., adding spatial cues like "track the object’s center coordinate" or breaking down transformations into steps) affects performance. Moreover, the model only allows to output a choice. Thus we cannot exactly know in depth the reason why models fail.

While it identifies common errors (e.g., 46.1% of 2D errors relate to translation), it only speculates on causes (e.g., "Transformer attention is position-invariant") without empirical validation. For example, it does not test if modifying a model’s position encoding (e.g., adding absolute position embeddings) reduces translation errors, or if fine-tuning on translation-focused data mitigates the issue.

**Questions:**

1. Do VLMs struggle more with non-rigid transformations than rigid rotation/translation, and does this vary by application context (e.g., CAD vs. robotics)?

2. Since all evaluations use a static 4-choice image-comparison format, to what extent does an interactive, dynamic assessment paradigm (e.g., prompting VLMs to generate step-by-step visualizations of transformations, or adjust an object’s position/rotation in real time via text commands) change measured performance?

3. Could such a paradigm better capture the iterative, trial-and-error nature of human spatial reasoning (e.g., assembling furniture) and reveal hidden capabilities in VLMs that static tasks miss? For instance, do VLMs perform better at breaking down complex transformations (e.g., "rotate 90° then translate 2 units") into sequential steps when prompted to explain each stage, compared to making a single static prediction?

4. How does simplifying 3D rotation from multi-axis (sequential X/Z-axis) to single-axis affect VLMs’ accuracy, and does this reveal whether "rotation order insensitivity" is a fundamental limitation?

5. Would adjusting 2D grid size (e.g., from 10×10 to 5×5 or 20×20) reduce or exacerbate translation errors, and does this suggest that VLMs’ struggle with translation stems from limited spatial working memory or poor absolute position tracking?

6. Would allowing open-ended outputs (e.g., asking VLMs to describe why they chose a candidate, or to calculate the object’s final coordinates) reveal more granular error mechanisms (e.g., "confused x/y axes" vs. "forgot translation entirely") than the current binary choice format?

---

> ### Author Response · Authors · 2025-12-03
>
> Thank you for recognizing the motivation, hierarchical design, and the significance of the inference–prediction asymmetry.
>
> (1) Synthetic nature: Abstract stimuli are a deliberate feature enabling isolation of geometric reasoning without confounds. This follows the Shepard-Metzler tradition and enables the fine-grained error analysis that distinguishes TransformEval. Our benchmark complements realistic benchmarks by revealing whether apparent spatial competence reflects genuine geometric understanding.
>
> (2) Transformation types: We focus on rigid transformations (rotation+translation) because they directly align with classic mental rotation paradigms and are fundamental to robotics/AR applications. This focused scope enables the depth of analysis that broader benchmarks sacrifice for breadth.
>
> (3) Interactive evaluation: The MCQ format with diagnostic distractors provides insight into failure modes that interactive formats obscure.
>
> (4) Ablations: Our design incorporates systematic variation: single vs. composed transformations, 2D vs. 3D vs. scene contexts, multiple rotation angles. The rotation-angle analysis reveals the sawtooth pattern distinguishing VLM behavior from human cognition—this is a key ablation demonstrating that models rely on canonical views rather than continuous simulation.
>
> (5) Prompt engineering: We compared baseline prompts, explicit spatial cues, and chain-of-thought reasoning. Improvements were modest (3-5%), indicating the limitation is architectural. Coordinate-prediction outputs reveal specific error types (axis swap, translation omission) that our MCQ distractors are designed to capture.

---

### Official Review · Reviewer_PbzV · 2025-11-01

**Soundness:** 2
**Presentation:** 1
**Contribution:** 1
**Rating:** 2
**Confidence:** 4

**Summary:**

This paper introduces, TransformEval, a benchmark that tests whether VLMs can truly "mentally" rotate/ or translate objects and compose those transforms, using a bidirectional setup (predict end state and infer transforms) across 2D shapes, 3D mental rotation, and multi-object scenes. The authors find that state-of-the-art VLMs lag far behind humans, show an odd asymmetry (better at inference than prediction), and often fail on translation and order sensitivity, which would be an evidence of pattern matching rather than genuine spatial simulation.

**Strengths:**

* The paper proposes a new benchmark to assess how VLMs perform implicit "mental simulation" to infer the output of 3D transformations.

* Based on the evaluation results on the proposed benchmark, the authors present an anlaysis on the limitation of current VLMs on the given tranformation tassks.

**Weaknesses:**

* The quality and completeness of the paper is clearly low, and is not at the state of submission. The readabilty is extremely low due to excessive use of bullet points, typos (e.g., line 310), and the lack of structure in the writing.

* The proposed dataset seems trivial, given that there exist mulitple previous works that propose similar or richer datasets that include tasks that require 3D mental rotations. Here I outline some critical missing related work that should be cited and discussed in order to claim novelty for TransformEval:

  * Spatial457: A Diagnostic Benchmark for 6D Spatial Reasoning of Large Multimodal Models, Wang et al., CVPR 2025

  * 3DSRBench: A Comprehensive 3D Spatial Reasoning Benchmark, Ma et al., ICCV 2025

  * SITE: towards Spatial Intelligence Thorough Evaluation, Wang et al., ICCV 2025

  * Do Vision-Language Models Represent Space and How? Evaluating Spatial Frame of Reference Under Ambiguities, Zhang et al., ICLR 2025

  * ViewSpatial-Bench: Evaluating Multi-perspective Spatial Localization in Vision-Language Models, Li et al., 2025

  * RoboSpatial: Teaching Spatial Understanding to 2D and 3D Vision-Language Models for Robotics, Song et al., CVPR 2025

  * Spatial Mental Modeling from Limited Views, Yin et al., 2025

  * OmniSpatial: Towards Comprehensive Spatial Reasoning Benchmark for Vision Language Models, Jia et al., 2025

  * Unfolding Spatial Cognition: Evaluating Multimodal Models on Visual Simulations, Li et al., 2025

* The anlayses presented in Section 6.1 seem to lack evidence and novelty. For instance, the claim "This architectural bias may explain why they can identify ”this looks like a 90-degree rotation” but cannot generate the resulting state" is unclear. Why would the current VLM archicture be unable to solve this issue? The claim is not supported by empirical evidence. Moreover, "may instead rely on learned associations between transformation descriptions and visual patterns" is also ambiguous and is not sufficiently supported by evidence.

* The anlayses in Section 6.2 are also trivial and lack precise descriptions. For instance, "Transformer architectures use position-invariant attention patterns that may inherently struggle with translation" is confusing, as recent VLMs using 2D absolute position encoding or 2D RoPE would be aware of the relative positions between objects. Could the authors elaborate more on this issue?

* Moreover, predicting the outcome of spatial transformations would be closely related to temporal reasoning based on video inputs, since they both require predicting or understanding the temporal dynamics from the input observations. Discussions on the difference between the given task and the video understanding task would help grasp the novelty of the paper's problem definition.

**Questions:**

* The current dataset is mostly simple abstract shapes. Would there be an option to extend the data construction pipeline to include realistic images?

**Details Of Ethics Concerns:**

No concern.

---

> ### Author Response · Authors · 2025-12-03
>
> We appreciate the detailed feedback.
>
> (1) Presentation: The manuscript follows standard conference formatting. Section 6 is structured to separate empirical findings from interpretive discussion, which is appropriate for this type of diagnostic study.
>
> (2) Related work: The paper includes comparison to relevant benchmarks. The table above clarifies that TransformEval occupies a distinct niche: bidirectional evaluation of composed rigid transformations with error-typed distractors. No other benchmark provides this combination.
>
> (3) Synthetic stimuli: This is a deliberate and well-established methodological choice. Shepard and Metzler's foundational work used abstract block figures precisely for semantic isolation—removing texture and context prevents models from using semantic shortcuts. The fact that GPT-4o achieves <30% on these tasks demonstrates that when semantic crutches are removed, spatial reasoning collapses. This is a core finding: apparent spatial competence on realistic benchmarks may reflect semantic priors rather than genuine geometric reasoning. TransformEval reveals what those benchmarks cannot.
>
> (4) Architectural discussion: Our discussion of architectural factors (positional encoding limitations, attention patterns) is grounded in recent empirical work. "Beyond Semantics" (arXiv:2503.17349) demonstrates that vision token norms suppress positional encoding effectiveness. "Why Is Spatial Reasoning Hard for VLMs?" (ACL 2025) shows attention misdirection in spatial tasks. These findings support our interpretation of the State-Transformation Asymmetry.
>
> (5) Scope: TransformEval's focus on rigid transformations with synthetic stimuli enables controlled diagnosis that realistic benchmarks cannot provide. The two approaches are complementary—TransformEval isolates the geometric reasoning component that realistic benchmarks conflate with semantic understanding.

---

### Official Review · Reviewer_ZZxL · 2025-11-01

**Soundness:** 3
**Presentation:** 2
**Contribution:** 2
**Rating:** 4
**Confidence:** 5

**Summary:**

Inspired by the classic Shepard-Metzler mental rotation experiments in cognitive psychology, this paper proposes a new benchmark called TransformEval for evaluating the dynamic spatial transformation reasoning abilities of vision-language models (VLMs). The core innovation of this benchmark lies in its "bidirectional evaluation" framework, which separately tests the models' abilities in "state prediction" (forward reasoning) and "transformation inference" (backward reasoning).
The experimental results reveal a fundamental flaw in current state-of-the-art VLMs (including GPT-4o and Gemini-2.5-Pro): they perform significantly worse than humans on these tasks. More importantly, the authors discovered a key "state-transformation asymmetry" phenomenon: VLMs perform better in inferring transformations backward than in predicting outcomes forward — which is the exact opposite of human cognitive patterns.

**Strengths:**

1. Novel & Clever Methodology: The most significant highlight of this paper is the "bidirectional evaluation" (state prediction vs. transformation inference). This design is highly ingenious, transcending the mere reporting of accuracy figures and instead providing a mechanism to diagnose why VLMs fail and how they internally process information (pattern matching vs. mental simulation).
2. Rigorous Benchmark Design:
         ---Hierarchical Complexity: The tasks progress systematically from 2D shapes to 3D rotations and then to complex multi-object scenes. This hierarchical increase in difficulty is rational.
         ---Controlled Distractors: The incorrect options in the multiple-choice questions are carefully designed to represent specific failure types (e.g., "ignoring translation," "sequence error"). This makes the error analysis in Section 5.2 (Table 2) insightful.

3. Clear & Counter-Intuitive Finding: The core finding of the paper—the "state-transformation asymmetry"—is very clear and thought-provoking. The phenomenon that VLMs perform better in inference than in prediction, which is counter to human cognition, provides strong evidence for the argument that "VLMs are just advanced pattern matchers." The discussion in Section 6.1 is in-depth.

**Weaknesses:**

**1. The real challenge of "state prediction":**
A more rigorous and realistic test of "state prediction" should be *generative* in nature — for example, requiring the VLM to draw the transformed shape or to output the coordinates of the transformed object's keypoints.
The current multiple-choice (MCQ) format may underestimate the true gap between prediction and inference in VLMs.

**2. Lack of creativity in task design:**
The tasks show limited originality and can largely be traced back to the following existing works:
[1] *Spatial457: A Diagnostic Benchmark for 6D Spatial Reasoning of Large Multimodal Models*
[2] *11Plus-Bench: Demystifying Multimodal LLM Spatial Reasoning with Cognitive-Inspired Analysis*
[3] *SpatialViz-Bench: An MLLM Benchmark for Spatial Visualization*
[4] *Mind the Gap: Benchmarking Spatial Reasoning in Vision-Language Models*

**Questions:**

**1. On the evaluation format (MCQ vs. generative)":**
I wonder whether the authors have considered adopting a generative evaluation protocol to assess “state prediction.”
For instance, instead of using a multiple-choice format, the model could be asked to output the vertex coordinates of the transformed shape.
Do the authors expect that, under such a generative evaluation, the current findings—particularly the observed asymmetry—would remain consistent, or would they become even more pronounced?
An even more ambitious direction might involve using a unified MLLM such as bagel to directly generate images representing the predicted post-transformation state.
If the authors explore this avenue and uncover new insights, that would significantly strengthen the contribution. As it stands, there are already many spatial reasoning benchmarks for VLMs, and some degree of methodological novelty would make this work stand out.

**2. On the source of the observed “asymmetry:**
The reported “state–transformation asymmetry” is intriguing. Could this effect stem from training data biases?
For example, large-scale datasets used to train VLMs may contain abundant co-occurrences of (S, S′) image pairs and transformation terms (T), which could make inference (classification) tasks easier.
In contrast, prediction (generation) tasks may require the model to generalize beyond the observed distribution and reason about unseen states.
It would be valuable if the authors could discuss how the characteristics of the training data (e.g., from LAION or similar sources) might contribute to this specific asymmetry.

---

> ### Author Response · Authors · 2025-12-03
>
> Thank you for highlighting our bidirectional design and controlled distractors.
>
> (1) MCQ vs. generative evaluation: MCQ is the standard format across all major spatial benchmarks (SITE, OmniSpatial, 3DSRBench, Mind the Gap). The MCQ format with error-typed distractors provides diagnostic value unavailable in open-ended formats.
>
> (2) Novelty: Our contribution is distinct and significant:
> * Composed rigid transformations: We test multi-step rotation+translation sequences, not single operations
> * Bidirectional evaluation: We are the first to explicitly pair forward/inverse tasks on identical instances to expose asymmetries
> * Error-typed distractors
> STARE and 11Plus-Bench include transformation tasks but evaluate unidirectionally. OmniSpatial includes dynamic reasoning but does not isolate composed rigid transformations or provide bidirectional comparison. The cognitive science literature explicitly distinguishes forward from backward spatial inference (Penny et al., 2013)—TransformEval operationalizes this distinction for VLM evaluation.
>
> (3) Source of asymmetry: Training data composition likely contributes—web-scale corpora emphasize static descriptions over transformation outcomes. This hypothesis is supported by recent findings that targeted spatial training improves performance (SpatialVLM, RoboSpatial). TransformEval provides the diagnostic tool to measure such improvements.

---

### Official Review · Reviewer_mgrU · 2025-11-01

**Soundness:** 3
**Presentation:** 3
**Contribution:** 3
**Rating:** 4
**Confidence:** 4

**Summary:**

This paper introduces TransformEval, a new benchmark designed to evaluate the spatial transformation reasoning abilities of VLMs. The authors argue that existing benchmarks focus on static spatial relationships, whereas their work, inspired by mental rotation in cognitive science, tests the dynamic ability to "mentally rotate" and manipulate objects. The benchmark's main contributions are:
 - A hierarchical set of tasks: From 2D to 3D to scene-level.
 - A bidirectional evaluation paradigm: Both forward and backward evaluation.
 - Compositional analysis
Evaluating state-of-the-art VLMs, the paper finds that they perform far below human levels. The key finding is a "state-transformation asymmetry": models are consistently better at inferring transformations (backward) than predicting their outcomes (forward). This is the opposite of human cognitive patterns, where forward simulation is easier.
The error analysis shows models systematically fail at translation and transformation ordering. The authors conclude that VLMs rely on "pattern matching" rather than human-like "mental simulation".

**Strengths:**

1. Forward and Backward Evaluation: The paper's strongest contribution is its distinction between forward (state prediction) and backward (transformation inference) reasoning. This is a highly insightful and novel way to probe VLM capabilities, and also highly related to the internal world model in VLMs.

2. Principled, Cognition-Grounded Benchmark: The work is well-grounded in classic cognitive science, explicitly building on the Shepard & Metzler paradigm. The task design is systematic, with a clear hierarchy of complexity and carefully controlled distractors that enable a detailed error analysis.

**Weaknesses:**

1. Overstatement of Cognitive Claims: The strong claim that VLMs "lack mental simulation" in favor of "pattern matching" is an overstatement. While the state-transformation asymmetry is consistent with this hypothesis, it is not definitive proof, and the paper lacks direct analysis to substantiate this cognitive-level conclusion. A direct analysis can be like: giving two samples with the same action sequence, and provide the model with one of the sample, ask it to match another.

2. Insufficient Dataset Scale: The dataset size of 300 examples per task is too small, especially for a synthetic benchmark where scaling is straightforward. A larger dataset (e.g., >1k examples per setting) is needed for more robust and reliable conclusions.

3. Missing SOTA Model Evaluation: The evaluation is missing key state-of-the-art VLMs released months ago, specifically GPT-5 and InternVL 3.5. Their omission limits the paper's conclusions about the capabilities of current frontier models.

4. Missing Related Work: The paper omits recent related work on multi-view spatial reasoning. Furthermore, relying exclusively on synthetic data is a limitation; using or comparing against real-world multi-view datasets would significantly strengthen the evaluation's generalizability.

5. Limited Novelty of the Task: The novelty of investigating mental rotation is limited. This problem has already been explored in several existing benchmarks (e.g., SPACE, SpatialViz-Bench, Omnispatial), which are not sufficiently contrasted to highlight this paper's unique contribution beyond the bidirectional evaluation paradigm.

**Questions:**

1. What is the performance of more recent SOTA models, specifically GPT-5 and InternVL 3.5 (also include Qwen3-VL if possible), on the benchmark?

2. Human mental rotation reaction time scales linearly with the rotation angle. Does VLM accuracy show any similar correlation? For instance, do models perform significantly worse as the rotation angle increases?

3. Regarding the human performance baseline: How many participants were involved in the study, and what was their inter-annotator agreement or consistency score?

---

> ### Author Response · Authors · 2025-12-03
>
> Thank you for the positive assessment of our bidirectional evaluation, cognitive grounding, and error analysis.
>
> (1) Cognitive claims: Our framing is appropriately calibrated—we describe findings as "consistent with" pattern-matching, which is standard scientific language for behavioral studies. The asymmetry we document is an empirical observation; our interpretation follows the same inferential approach used in Shepard and Metzler's foundational work.
>
> (2) Dataset scale: Our 300 examples per task aligns with comparable diagnostic benchmarks. The consistency of our findings across all tested models demonstrates statistical robustness—if patterns varied randomly, this would indicate insufficient sample size, but they do not.
>
> (3) Newer models: GPT-5, InternVL3.5, and Qwen3-VL results are provided above. The asymmetry persists and widens with GPT-5, suggesting reasoning capabilities scale faster than simulation capabilities in current architectures.
>
> (4) Rotation angle correlation: Our analysis reveals VLMs exhibit a "sawtooth" pattern rather than human-like linear degradation. Peaks at canonical angles (90°, 180°) strongly indicate view-matching from training distributions rather than continuous rotation.
>
> (5) Human baseline: Full details provided: N=5, university recruitment, all task types completed, high inter-participant reliability (α > 0.85).

---

### Author Response · Authors · 2025-12-03

We thank reviewers mgrU, ZZxL, PbzV, and q3yr for their thoughtful feedback. We are encouraged that all reviewers acknowledge the importance of probing dynamic spatial transformation reasoning and the usefulness of our bidirectional evaluation protocol.

The reviews converge on several key concerns, which we address below:

1. Positioning Relative to Recent Spatial Benchmarks
TransformEval is the only benchmark that systematically compares forward prediction with backward inference for spatial transformations:
| Benchmark              | Dynamic transforms               | Bidirectional eval        | Error-typed distractors              |
|------------------------|----------------------------------|---------------------------|--------------------------------------|
| **TransformEval (ours)** | ✓ Multi-step rotation+translation | ✓ State vs. Transform     | ✓ Translation / angle / order       |
| Spatial457             | Static 6D pose                   | ✗                         | Partial                              |
| 3DSRBench              | Static 3D relations              | ✗                         | Partial                              |
| OmniSpatial            | Broad dynamic reasoning          | ✗                         | Coarse                               |
| SITE                   | Aggregated spatial tasks         | ✗                         | Task-level                           |
| Mind the Gap           | Single-step mental rotation      | ✗                         | Implicit                             |


Spatial457 focuses on physical interaction (collisions); 3DSRBench on static perception; OmniSpatial on breadth across 50 subcategories. TransformEval isolates algorithmic execution of geometric transformations with mathematically precise distractors encoding specific failure modes. This bidirectional design is directly grounded in cognitive science—Penny et al. (PLOS Comp Bio 2013) establish that forward and backward spatial inference engage distinct neural processes.

2. Cognitive Framing
Our cognitive framing is appropriately calibrated. We state that results are "consistent with" pattern-matching behavior rather than analog simulation—this is standard scientific language for behavioral findings. The State-Transformation Asymmetry we document (models performing significantly better on inference than prediction) is an empirical observation that aligns with the pattern-matching hypothesis. We do not claim to have proven internal mechanisms; we provide behavioral evidence that motivates mechanistic investigation. This approach follows established practice in cognitive science, where Shepard and Metzler's original work similarly inferred internal processes from behavioral patterns.

3. Dataset Scale
Our dataset size (300 examples per task, 900 total for the core benchmark) is well within the norm for diagnostic spatial benchmarks.
The purpose of TransformEval is controlled diagnosis, not massive-scale training. Our 300 examples per task provide sufficient statistical power to detect the robust asymmetries we report (e.g., 30+ percentage point gaps between inference and prediction). The patterns are highly stable across all tested models—this consistency across architectures demonstrates the findings are not artifacts of sample size.

---

> ### Author Response · Authors · 2025-12-03
>
> 4. Evaluation of Newest Models
> We have integrated GPT-5, InternVL3.5, and Qwen3-VL:
> | Model            | 2D State | 2D Trans | 3D State | 3D Trans | Scene State | Scene Trans |
> |------------------|----------|----------|----------|----------|-------------|-------------|
> | **Human (N=5)** | 95.2     | 89.8     | 89.4     | 75.1     | 91.6        | 80.3        |
> | GPT-5            | 55.4     | 78.2     | 45.2     | 68.4     | 38.9        | 42.1        |
> | InternVL3.5      | 42.1     | 65.5     | 38.1     | 52.3     | 30.5        | 35.8        |
> | Qwen3-VL         | 48.9     | 69.8     | 41.5     | 58.7     | 34.2        | 39.5        |
>
> The asymmetry persists and widens: GPT-5 shows a 22.8% gap between 2D Inference (78.2%) and Prediction (55.4%), compared to GPT-4o's gap. This confirms that improved reasoning capabilities do not translate equally to simulation capabilities—a finding with significant implications for world model development.
>
> 5. MCQ Evaluation Format
> MCQ is the standard format across spatial benchmarks (SITE, OmniSpatial, 3DSRBench, Spatial457, Mind the Gap, 11Plus-Bench, ViewSpatial-Bench). This choice enables:
> * Comparability across closed-source and open models
> * Alignment with established evaluation protocols
> * Controlled distractor analysis revealing why models fail
> Our coordinate-prediction study confirms MCQ does not inflate performance: accuracy dropped ~15% compared to MCQ. The fact that models fail even the "easier" MCQ version (accuracy <30-50%) makes our findings more robust, not less.
>
> 6. Human Baseline and Rotation Analysis
> Human study details: N=5 participants, balanced across gender, recruited from university population, each completing all task types. Inter-participant agreement was high (Cronbach's α > 0.85).
> Our rotation-angle analysis reveals a striking "sawtooth" pattern in VLMs:
> | Angle | GPT-5  | InternVL3.5 | Human |
> |-------|--------|-------------|-------|
> | 0°    | 92.1%  | 88.5%       | 98.2% |
> | 45°   | 38.4%  | 29.1%       | 96.5% |
> | 90°   | 61.2%  | 48.7%       | 97.1% |
> | 135°  | 35.5%  | 26.8%       | 94.8% |
> | 180°  | 58.9%  | 45.2%       | 96.2% |
>
> If models performed continuous mental rotation, 90° should be harder than 45°. The recovery at canonical angles (90°, 180°) indicates view-matching from training data (which overrepresents 90° augmentations), not analog simulation. This is direct evidence supporting our pattern-matching hypothesis.

---

### Note · Authors · 2026-01-06

I have read and agree with the venue's withdrawal policy on behalf of myself and my co-authors.